# Remote Photoplethysmography and Motion Tracking Convolutional Neural Network with Bidirectional Long Short-Term Memory: Non-Invasive Fatigue Detection Method Based on Multi-Modal Fusion

**Lingjian Kong** [1] , **Kai Xie** [1,*], **Kaixuan Niu** [1], **Jianbiao He** [2] **and Wei Zhang** [3]

[1] School of Electronic Information and Electrical Engineering, Yangtze University, Jingzhou 434023, China; 202003899@yangtzeu.edu.cn (L.K.); 202101374@yangtzeu.edu.cn (K.N.)
[2] School of Computer Science, Central South University, Changsha 410083, China; jbhe@mail.csu.edu.cn
[3] School of Electronic Information, Central South University, Changsha 410083, China; csuzwzbn@csu.edu.cn
* Correspondence: xiekai@yangtzeu.edu.cn; Tel.: +86-136-9731-5482

**Abstract:** Existing vision-based fatigue detection methods commonly utilize RGB cameras to extract facial and physiological features for monitoring driver fatigue. These features often include single indicators such as eyelid movement, yawning frequency, and heart rate. However, the accuracy of RGB cameras can be affected by factors like varying lighting conditions and motion. To address these challenges, we propose a non-invasive method for multi-modal fusion fatigue detection called RPPMT-CNN-BiLSTM. This method incorporates a feature extraction enhancement module based on the improved Pan–Tompkins algorithm and 1D-MTCNN. This enhances the accuracy of heart rate signal extraction and eyelid features. Furthermore, we use one-dimensional neural networks to construct two models based on heart rate and PERCLOS values, forming a fatigue detection model. To enhance the robustness and accuracy of fatigue detection, the trained model data results are input into the BiLSTM network. This generates a time-fitting relationship between the data extracted from the CNN, allowing for effective dynamic modeling and achieving multi-modal fusion fatigue detection. Numerous experiments validate the effectiveness of the proposed method, achieving an accuracy of 98.2% on the self-made MDAD (Multi-Modal Driver Alertness Dataset). This underscores the feasibility of the algorithm. In comparison with traditional methods, our approach demonstrates higher accuracy and positively contributes to maintaining traffic safety, thereby advancing the field of smart transportation.

**Keywords:** intelligent traffic; fatigue detection; multi-modal feature fusion; heart rate; bidirectional LSTM

## 1. Introduction

Recently, with the rapid expansion of the transportation industry and the widespread use of vehicles, instances of traffic accidents resulting from fatigue driving have become increasingly common. Prolonged periods of driving or insufficient sleep can induce fatigue in drivers, significantly elevating the risk of accidents. Research indicates that driver drowsiness and sleep deprivation are primary contributors to road traffic accidents [1], accounting for approximately 25% to 30% of such incidents [2]. The ramifications of traffic accidents extend beyond individual safety and property loss, permeating into the broader stability of both a nation and society. Consequently, the timely detection of fatigue in drivers and the provision of alerts to prompt breaks are critical measures for upholding traffic safety and ensuring secure travel. In light of these considerations, addressing the issue of fatigue-related accidents assumes paramount importance. Developing effective methods for detecting and mitigating driver fatigue can substantially contribute to reducing accident rates, thereby safeguarding lives, property, and the overall stability of society.

Currently, fatigue driving detection methods can be broadly categorized into advantage detection, single-mode feature detection, and multi-mode feature detection [3]. Advantages are primarily assessed with public questionnaires and advantage scales. However, these individual methods exhibit significant differences, and their time-consuming nature renders them insufficient for real-time detection and prevention. This article addresses fatigue characteristics in two other dimensions.

Single-modal feature detection relies on individual features to assess fatigue. Among the current methods focusing on single-modal features, utilizing facial features has proven to be effective in determining a driver's fatigue status. Facial features encompass expressions, eye states, head posture, etc., extracted from a driver's facial images or videos. For instance, Zhuang [4] introduced an efficient fatigue detection method based on eye status, utilizing pupil and iris segmentation. Yang et al. [5] proposed a yawn detection method based on subtle facial action recognition, utilizing 3D convolution and bidirectional long short-term memory networks to detect a driver's fatigue state. Liu [6] presented a fatigue detection algorithm based on facial expression analysis. Xing [7] applied a convolutional neural network to face recognition, implementing a straightforward eye state judgment method using the PERCLOS algorithm to determine a driver's fatigue state, with experimental results demonstrating an 87.5% fatigue recognition rate. Moujahid [8] introduced a face monitoring system based on compact facial texture descriptors, capable of encompassing the most discriminative drowsy features. Bai [9] utilized the facial landmark detection method to extract a driver's facial landmarks from real-time videos, subsequently obtaining driver drowsiness detection results with 2s-STGCN and significantly improving driver drowsiness detection. Ahmed [10] proposed an ensemble deep learning architecture operating on merged features of eye and mouth subsamples, along with decision structures, to ascertain driver fitness.

The exploration of fatigue driving detection based on physiological characteristics has evolved into a significant research direction. In recent years, traditional heart rate detection has predominantly relied on wearable devices utilizing electroencephalogram (EEG) or electrocardiogram (ECG). For instance, Zhu [11] proposed a wearable EEG-based vehicle driver drowsiness detection method using a convolutional neural network (CNN). Gao [12] developed a novel EEG-based spatiotemporal convolutional neural network (ESTCNN) for driver fatigue detection, achieving a high classification accuracy of 97.37%. Despite their accuracy, traditional methods are hindered by issues such as expensive equipment and inconvenient wearing. In response to these challenges, non-contact physiological feature extraction has emerged as a research hotspot. Heart rate (HR) and heart rate variability (HRV) are crucial vital signs, with their changes directly or indirectly reflecting information on the physiological state of the human body. Research indicates that heart rate and HRV can objectively indicate fatigue. For instance, Dobbs [13] used a portable device to conveniently record HRV, showing a small absolute error compared with electrocardiography. Monitoring changes in heart rate and HRV is crucial for determining driver fatigue. Lu [14] emphasized the significance of HRV as a physiological marker for detecting driver fatigue, measurable during real-life driving. Systematic reviews, such as the one conducted by Persson [15], explore the relationship between HRV measurements and driver fatigue, as well as the performance of HRV-based fatigue detection systems. In medical contexts, Allado et al. [16] evaluated the accuracy of imaging photoplethysmography compared to existing contact point measurement methods in clinical settings, demonstrating that rPPG can accurately and reliably assess heart rate. Cao [17] et al. introduced a drowsiness detection system using low-cost photoplethysmography (PPG) sensors and motion sensors integrated into wrist-worn devices. Comas [18] proposed a lightweight neural model for remote heart rate estimation, focusing on efficient spatiotemporal learning of facial photoplethysmography (PPG). Patel [19] introduced an artificial intelligence-based system designed to detect early driver fatigue by leveraging heart rate variability (HRV) as a physiological measurement. Experimental results demonstrated that this HRV-based fatigue detection technology served as an effective countermeasure against fatigue. Gao [20]

proposed a novel remote heart estimation algorithm incorporating a signal quality attention mechanism and a long short-term memory (LSTM) network. Experiments indicated that the LSTM with an attention mechanism accurately estimated heart rate from corrupted rPPG signals, performing well across cross-subject and cross-dataset tasks. Additionally, the signal quality model's predicted scores were found to be valuable for extracting reliable heart rates. The accuracy of existing heart rate detection based on RGB cameras is susceptible to various factors such as lighting conditions and motion, leading to challenges in achieving precise heart rate estimation. Recent advancements have addressed these challenges. For instance, Yin [21] and colleagues proposed a new multi-task learning model combining the strengths of signal-based methods and deep learning methods to achieve accurate heart rate estimation, even in scenarios with changing lighting and head movement. Given the dynamic lighting changes typical in vehicle cabins, heart rate measurement in automotive contexts presents specific challenges. To tackle these issues, Ming [22] and collaborators introduced a method named Illumination Variation Robust Remote Photoplethysmography (Ivrr-PPG) for monitoring a driver's heart rate during road driving. Rao [23] proposed a distracted driving recognition method based on a deep convolutional neural network using in-vehicle camera-captured driving image data. Experimental analysis indicated an accuracy of 97.31%, surpassing existing machine learning algorithms. Consequently, methods based on deep convolutional neural networks prove effective in enhancing the accuracy of distracted driving identification. Addressing challenges related to dramatic lighting changes and significant driver head movements during driving, Nowara [24] demonstrated that narrowband near-infrared (NIR) video recordings can mitigate external light variations and yield reliable heart rate estimates. Rajesh [25] utilized the Pan–Tompkins method for R-peak detection to identify irregularities in human heart rate (DIIHR), achieving an average accuracy of 96.

The above-mentioned fatigue driving detection methods mainly rely on single modal data, which limits the adaptability to various scenarios and the reliability of model processing. Each parameter has its advantages and disadvantages. Therefore, identifying how to effectively combine and utilize multiple driver characteristics is an important research direction for real-time and accurate driver detection.

There are currently some methods that combine multi-modal data together for fatigue detection, which involve multi-modal feature fusion models. Most of the existing multi-modal fusion is implemented based on decision fusion and feature fusion methods of RGB cameras. For example, Kassem [26] proposed a low-cost driver fatigue level prediction framework (DFLP) for detecting driver fatigue at the earliest stage. Experimental results show that this method can predict the driver fatigue level with an overall accuracy of 93.3%. Du [27] proposed a novel non-invasive method for driver multi-modal fusion fatigue detection by extracting eyelid features and heart rate signals from RGB videos. The results show that the multi-modal feature fusion method can significantly improve the accuracy of fatigue detection. Dua and colleagues [28] proposed a driver drowsiness detection system in their paper. They use the driver's RGB video as input to help detect drowsiness. The results show that the accuracy of the system reaches 85%. Liu [29] focused on RGB-D cameras and deep learning generative adversarial networks and utilized multi-channel schemes to improve fatigue detection performance. Research indicates that fatigue features extracted with convolutional neural networks outperform traditional manual fatigue features. However, relying on a single feature may not guarantee robustness. Du [30] and colleagues used a single RGB-D camera to extract three fatigue features: heart rate, eye-opening, and mouth-opening. They proposed a novel multi-modal fusion recurrent neural network (MFRNN) that integrates these three features to enhance the accuracy of driver fatigue detection. To address issues such as poor comfort, susceptibility to external factors, and poor real-time performance in existing fatigue driving detection algorithms, Jia [31] designed a system for detecting driver facial features (FFD-System) and an algorithm for judging driver fatigue status (MF-Algorithm). Akrout [32] proposed a fusion system based on yawn detection, drowsiness detection, and 3D head pose estimation.

Traditional fatigue detection methods often require the connection of inconvenient sensors (such as EEG and ECG) or use video camera systems sensitive to light, compromising privacy. Akrout suggests accounting for changes in lighting conditions during the day and night to avoid limiting the fusion system. Using an infrared camera could be a potential solution. Zhang [33] introduced Ubi-Fatigue, a non-contact fatigue monitoring system combining vital signs and facial features to achieve reliable fatigue detection. The results demonstrated that Fatigue-Radio's detection accuracy reached 81.4%, surpassing ECG or visual fatigue detection systems. Ouzar [34] and colleagues compared the performance of a single-modal approach using facial expressions or physiological data with a multi-modal system fusing facial expressions with video-based physiological cues. The multi-modal fusion model improved emotion recognition accuracy, with the fusion of facial expression features and iPPG signals achieving the best accuracy of 71.90%. This underscores the efficacy of multi-modal fusion, particularly in combining facial expression features with iPPG signals for enhanced emotion recognition accuracy.

To summarize, the existing fatigue driving detection systems face limitations in equipment deployment, environmental changes, and real-time monitoring. Addressing these challenges represents a crucial research direction for the future development of driving fatigue detection systems [35]. Consequently, this article will concentrate on resolving the following three problems:

1. The problem of the low fatigue detection accuracy of a single feature. Traditional vision-based fatigue detection methods usually only use a single feature, such as facial features, physiological features, etc., resulting in low fatigue detection accuracy.
2. The problem of low feature extraction accuracy. Existing multi-modal fusion is mostly implemented based on RGB camera methods, and its detection accuracy will be affected by different lighting conditions, motion, etc., resulting in the inability to correctly detect a driver's fatigue state.
3. The problem of the poor robustness and temporal nature of detection models. In the actual driving environment, a driver's fatigue state changes dynamically, and the fatigue state is continuous time series data. The existing methods focus on processing the characteristics of a certain moment while ignoring the changes in fatigue characteristics over time, which affects the robustness of the detection model.

It can be seen that it is very important to design a multi-modal fatigue driving detection system with high accuracy, strong robustness, portability, and real-time performance.

To address the aforementioned challenges, we propose a non-invasive method for multi-modal fusion fatigue detection based on heart rate features and eye and face features. Our approach involves the use of an infrared camera in conjunction with rPPG and MTCNN to extract a driver's physiological features and eye and face features, respectively. This combination aims to reduce errors in extracting physiological signals and facial features caused by varying lighting conditions during the day and night. To enhance feature extraction accuracy, we implemented feature extraction enhancement modules based on an improved Pan–Tompkins algorithm and 1D-MTCNN. These modules aim to more accurately extract heart rate signals and eyelid features. Subsequently, we utilize one-dimensional convolutional neural networks (1D CNNs) to establish two models based on PERCLOS values and heart rate signals for fatigue detection. Heart rate signals and PERCLOS are critical analysis objects, and their accurate extraction is pivotal for driver fatigue detection. For the extraction of heart rate signals, we use singular spectrum analysis (SSA) and filtering technology to process rPPG physiological signals. This process aims to extract relatively pure heart rate signals and enhance detection accuracy. The heart rate signal is then analyzed in the time–frequency domain, and the time–frequency domain temporal feature matrix related to fatigue is extracted. This matrix is input into a one-dimensional convolutional neural network (1D CNN) to establish a fatigue detection model based on heart rate. For PERCLOS extraction, 1D-MTCNN is utilized to calculate the PERCLOS value. Specifically, the MTCNN algorithm is used for face detection and key point positioning, offering faster and more accurate results compared with traditional

algorithms while minimizing the impact of varying lighting conditions. Finally, the trained data results from the two models are input into the BiLSTM network, and the outputs of the two models are weighted to achieve multi-modal fusion fatigue detection.

## 2. Principles and Methods

This paper proposes a non-invasive method for multi-modal fusion fatigue detection based on heart rate features and eye and face features: RPPMT-CNN-BiLSTM. The overall framework of the multi-modal fusion fatigue driving detection model can be seen in Figure 1.

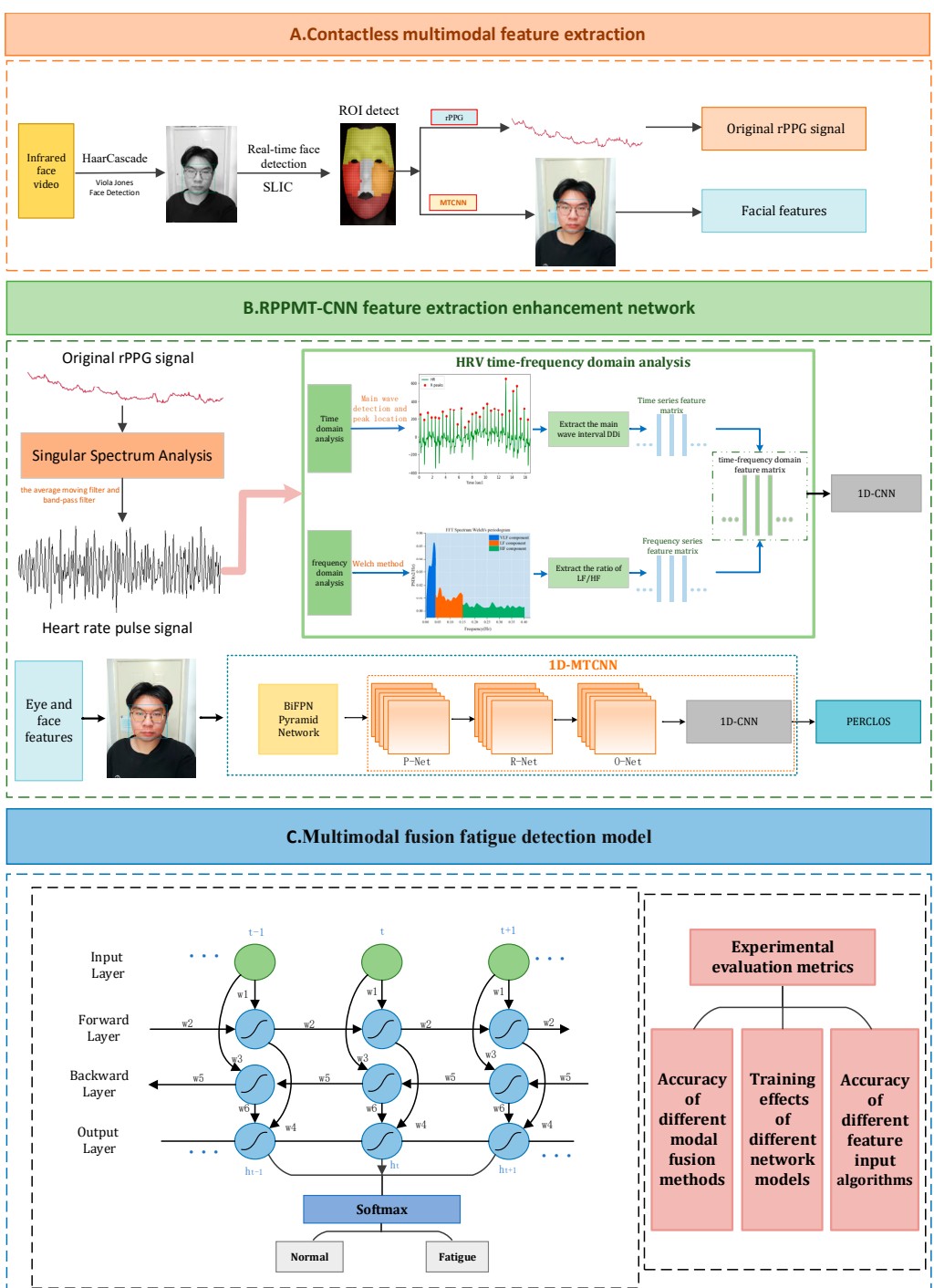

**Figure 1.** Multi-modal fusion fatigue driving detection model.

The model mainly includes the following 3 parts:

(1) Non-contact multi-modal feature extraction. We apply real-time face detection and ROI area tracking, utilizing infrared cameras in conjunction with rPPG and MTCNN combined with the BiFPN pyramid network to extract a driver's physiological characteristics and facial features, thereby reducing errors in extracting the driver's physiological signals and facial features caused by lighting changes during the day and night.

(2) An RPPMT-CNN feature extraction enhanced network. We introduce an infrared-based enhanced network for RPPMT-CNN feature extraction. In this paper, we establish an improved feature extraction enhancement module based on Pan–Tompkins and 1D-MTCNN. This module aims to extract heart rate signals and eyelid features more accurately. Subsequently, we create two fatigue detection models based on heart rate and PERCLOS values, utilizing one-dimensional convolutional neural networks (1D CNNs) for each model, respectively.

(3) A multi-modal feature fusion fatigue driving recognition model. To enhance the robustness and timeliness of fatigue detection, we introduce a multi-modal feature fusion fatigue driving recognition model. The outcomes of the trained model data are fed into the Bidirectional Long Short-Term Memory (BiLSTM) network. This allows the BiLSTM network to learn the temporal relationships between the data extracted from the 1D CNN, facilitating effective dynamic modeling of the input and output data. Ultimately, the outputs of the two models are weighted to achieve multi-modal fusion fatigue detection.

*2.1. Non-Contact Multi-Modal Feature Extraction*

2.1.1. Face Detection and ROI Area Tracking

When collecting a real-time driver video, accounting for the driver's head movement is crucial. Fixed-face Region of Interest (ROI) areas may inadvertently include non-skin areas alongside the actual skin area, thereby compromising the quality of subsequently extracted remote photoplethysmography (rPPG) signals. To address this, we use the Haar-Cascade face detector to identify faces in all frames of the video stream. Subsequently, we utilize the SLIC algorithm for superpixel skin segmentation on the detected face areas. This process determines the face ROI area for each frame in the picture, ensuring its precise position. The input video stream is segmented into multiple regions called superpixels. Superpixels corresponding to the cheek region, with the highest achromaticity in the forehead region, are selected as ROI. rPPG is then calculated for these selected superpixels, and the remaining superpixels are eliminated. This approach significantly reduces computation time. The method guarantees that during the extraction of physiological signals, the ROI area exclusively encompasses facial skin, thereby minimizing interference from motion artifacts.

2.1.2. Physiological Feature Extraction

Remote photoplethysmography (rPPG) is a non-contact method for extracting human physiological signals, developed based on the traditional photoplethysmography (PPG) principle. This approach leverages the periodic changes in blood flow induced by the human heartbeat within the skin capillaries, causing the absorption or reflection of periodic light signals. While these periodic signals are not directly observable by the human eye, high-definition cameras can capture facial data, enabling the analysis and monitoring of human physiological characteristics.

The advantage of rPPG technology lies in its non-invasive nature, as it eliminates the need for subjects to wear sensors, thereby avoiding interference with the human body. Additionally, the widespread availability and use of ordinary high-definition cameras have significantly reduced the cost of implementing rPPG technology, making it highly promising for various applications. For instance, in the context of driving fatigue monitoring, rPPG technology can be used to monitor a driver's heart rate and heart rate variability in real time. This real-time monitoring allows for the determination of the degree of fatigue,

enabling timely reminders for the driver to take necessary rest measures and ensuring overall driving safety.

For rPPG signal extraction, the approach involves calculating the average of the pixel intensity values within the Region of Interest (ROI) area. In each frame of the facial video, assumed to correspond to time $t$, all pixels within the infrared single channel in the selected ROI area are spatially averaged. The spatial average value of the ROI area at time $t$ can be expressed as:

$$a_t = \sum_{i=0}^{M} \frac{a_i}{M} \tag{1}$$

where is the total pixels in the selected ROI area and is the value of the $i - th$ pixel in the ROI area.

A 30s video (a total of 900 frames) is collected starting from time $t$ at a frame rate of 30 frames/second. The sequence of skin areas in consecutive image frames can be expressed as:

$$A_t = [a_t, a_{t+\tau}, a_{t+2\tau}, \cdots, a_{t+(N-1)\tau}], \ N = 900 \tag{2}$$

where is the time interval used to obtain one frame of video, which is the reciprocal of the sampling frequency.

The obtained signal is defined as the original input rPPG physiological signal at time t, and every 1 s (30 frames) thereafter, the original input rPPG physiological signal starting from the next second is obtained.

### 2.1.3. Facial Feature Extraction

To extract facial features from drivers in a fatigued state during driving, this article uses the MTCNN algorithm in conjunction with the BiFPN pyramid network as the core of the facial feature extraction module. The MTCNN algorithm comprises three cascaded networks (P-Net, R-Net, O-Net) and is utilized for face detection and key point localization. However, in complex driving environments, factors such as lighting changes, facial postures, gender, and partial occlusion may impact its performance. To enhance the algorithm's robustness, the BiFPN pyramid network is introduced, which better captures multi-scale features and improves adaptability to illumination changes. The output of BiFPN is connected with the cascade network of MTCNN to form a comprehensive facial feature extraction module. This approach yields a richer and more accurate representation of facial features. The algorithm demonstrates faster and more accurate performance than traditional methods, reducing the impact of varying lighting conditions. It maintains accurate face and key point detection even when a face is tilted, pitched, or partially obscured. Consequently, it is highly suitable for driver detection during driving. The structure of the BiFPN pyramid network is illustrated in Figure 2.

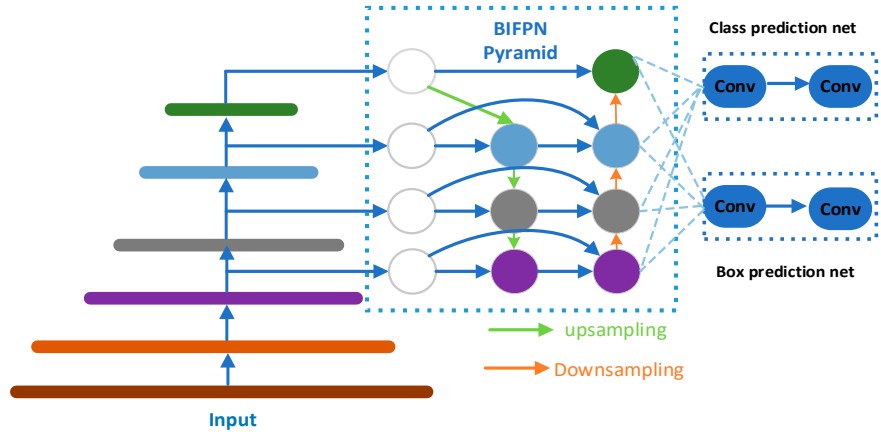

**Figure 2.** The BiFPN pyramid network.

The input $N$ videos are sampled at 30 frames/s to obtain n groups of frame images. The size of these images is reduced to 0.5 times the original images and formed into n sample sets. The sample set is used as the input of the MTCNN network and is calculated as follows:

$$L_i(\det, box, landmark) = \overline{MTCNN}(X_i) \tag{3}$$

where $i \in [1, n]$ $L_i(\det, box, landmark)$ represents the face candidate frames and key points $X_i$ obtained using the network output. Among them, $L_i(\det, box, landmark)$ includes the coordinate values of 5 key points on the face (left eye, right eye, nose, and left and right corners of the lips).

## 2.2. The RPPMT-CNN Feature Extraction Enhancement Network

After acquiring the initial facial information and original physiological signals, we incorporated the RPPMT-CNN feature extraction enhancement network. This method is grounded in an improved algorithm and devises a 1D CNN (convolutional neural network) structure tailored for facial feature processing, enabling the capture of spatiotemporal relationships inherent in facial features. Concurrently, a distinct 1D CNN structure was formulated for processing physiological features, aiming to more precisely capture the time–frequency domain characteristics of physiological signals. Following the separate optimization of facial and physiological features, their characteristic information can be maximally captured. The combination of facial and physiological information yields more accurate and comprehensive features, establishing the groundwork for subsequent comprehensive analysis and application.

### 2.2.1. Singular Spectrum Analysis

Due to the non-orthogonal characteristics of physiological sources, the usual blind source separation method cannot directly extract the heart rate pulse signal from the original rPPG signal. Therefore, based on singular spectrum analysis, we propose the following method to separate the target signal. The data matrix $A$ of each time series $A\_t$ of length $N$ can be expressed as:

$$A = \begin{pmatrix} a_t & a_{t+\tau} & \cdots & a_{t+(K-1)\tau} \\ a_{t+\tau} & a_{t+2\tau} & \cdots & a_{t+K\tau} \\ \vdots & \cdots & \ddots & \vdots \\ a_{t+(M-1)\tau} & a_{t+M\tau} & \cdots & a_{t+(N-1)\tau} \end{pmatrix} \tag{4}$$

where $K = N - M + 1$.

Then, we perform singular value decomposition (SVD) on the data matrix to solve the characteristic matrix of $A$. Its singular value decomposition expression is:

$$A = U\sum V^T \tag{5}$$

$$A = \sum_{i=1}^{M} U_i P_i^T \tag{6}$$

$$P_i = \sqrt{\lambda_i} V_i \tag{7}$$

where U and $V$ are the two orthogonal bases representing the left singular matrix and right singular matrix, respectively. The diagonal matrix $\sum$ is composed of singular values $\sigma_i$. It satisfies the relationship with the eigenvalue λ of AAT (covariance matrix) in eigenvalue decomposition (EVD): $\sigma_i = \sqrt{\lambda_i}$.

After singular value decomposition, the data matrix $A$ is decomposed into $M$ components. Then, we extract the heart rate pulse signal Ri from the M independent components,

where $R_i = P_i U_i^T (i < r)$. Finally, we recover the output time series $g_i(t)$ from $Ri$ using anti-angle averaging.

$$g_i(t) = \begin{cases} \frac{1}{m} \sum_{h=1}^{m} R_{h,t+1-h}^i, (t \leq K) \\ \frac{1}{m} \sum_{h=1}^{m} R_{t+h-K,K-h+1}^i, (t > K) \end{cases} \tag{8}$$

Due to significant noise corruption in the heart rate pulse signal obtained with singular spectrum analysis, further filtering is necessary. In this case, a moving average filter is used for low-pass filtering to eliminate low-frequency interference caused by factors such as breathing. The original sampled data forms a one-dimensional queue of length N, and a sliding window of length L is applied to it. The average value of the data within the window is computed as the output of the filter at the current moment. The window progresses in the positive direction of the time axis, generating filter outputs for subsequent moments until all the data points are covered. The calculation formula for the moving average filter at the $i - th$ moment is given by:

$$G(i) = \frac{1}{L} \sum_{j=1}^{j=i+L-1} g(j)(i = 1, 2, 3, \dots, N - L + 1) \tag{9}$$

Subsequently, a Hamming window bandpass filter with a passband frequency of 0.8~4 Hz is applied to eliminate high-frequency and low-frequency noise outside the heart rate range, aiming to minimize noise interference.

### 2.2.2. Time Domain Analysis of Heart Rate Signals

Building upon [10], this paper uses the enhanced Pan–Tompkins algorithm for primary wave detection and localization. The main wave detection involves a combination of Shannon energy and adaptive dual threshold methods to accurately identify the main wave and pinpoint its peak for extracting the target signal. The detailed algorithmic flow is illustrated in Figure 3.

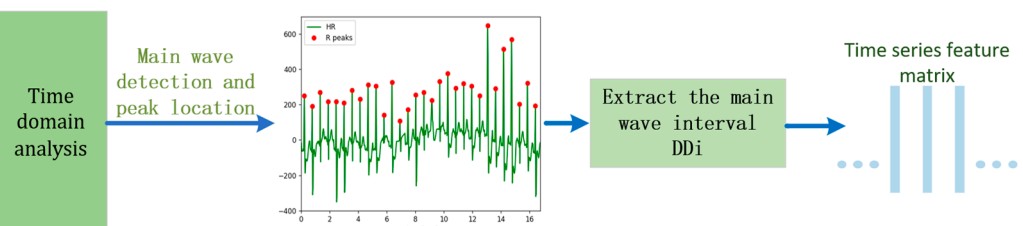

**Figure 3.** Time domain analysis of heart rate signals.

The time difference between two adjacent main wave peaks is called the DD interval, denoted as $DD_i(i = 1, 2, 3 \dots \dots)$. By integrating the physiological characteristics of the heart rate signal with pertinent medical insights, we designate the interval between the peaks of the main waves as the duration of one heartbeat, constituting a single cardiac cycle. According to existing research, the standard deviation of the cardiac cycle in the human body tends to notably increase as fatigue intensifies. Hence, this paper uses the standard deviation ($SD$) of the RR interval as the time domain analysis index, with its calculation formula as follows:

$$MEAN = \sum_{i=1}^{N} \frac{DD_i}{N} \tag{10}$$

$$SD = \sqrt{\frac{1}{N} \sum_{i=1}^{N} (DD_i - MEAN)^2} \tag{11}$$

The flow chart and specific implementation process of the improved algorithm are shown in Figure 4.

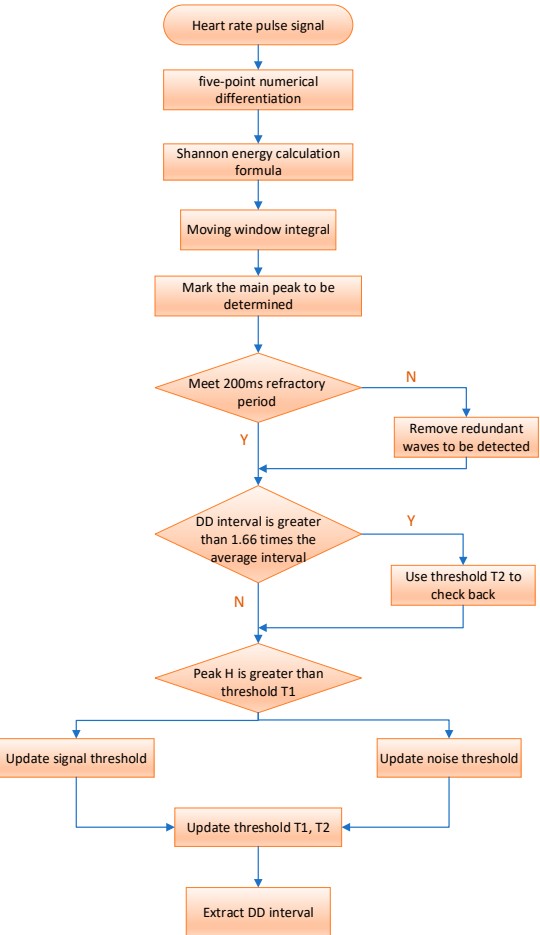

**Figure 4.** Flow of the HRV time domain analysis algorithm.

We initially differentiate the filtered heart rate pulse signal to extract waveform slope information, using the five-point numerical differentiation formula:

$$y\prime(i) = x[i - 2 : i + 2] \cdot [\frac{1}{12}, -\frac{8}{12}, 0, \frac{8}{12}, -\frac{1}{12}]^T \tag{12}$$

where $y'(i)$, $y'(i)$ represents the slope of the heart rate pulse signal at the $i - th$ time point, the symbol ":" is used to represent an array or vector, and the symbol "·" is used to represent matrix or vector multiplication. Before performing the Shannon energy calculation, the differentiated data are standardized as follows:

$$\widetilde{y(n)} = y(n)/\max_i(|y(n)|) \tag{13}$$

Then, the output of the derivative undergoes nonlinear amplification using the Shannon energy formula. This process ensures that all data points become positive, accentuates high- and medium-intensity components, and attenuates other intensity values. This enhancement aids in better locating the main wave and detecting its peak. The Shannon energy formula is as follows:

$$y(nT) = -[x(nT)]^2 \ln([x(nT)]^2) \tag{14}$$

Following the calculation of Shannon energy, numerous closely spaced and small wave peaks are obtained. To enhance the concentration of energy, a moving window integration is applied to smooth the waveform. The choice of window size is crucial for main wave detection. If the selected window is too small, the resulting signal waveform after moving

integration may lack smoothness, hindering main wave peak detection and potentially leading to false detections. Conversely, if the window is too large, the energy of the main wave in the signal may be dispersed, increasing the risk of missed detection.

Typically, the size of the moving window integral after Shannon energy processing is correlated with the sampling frequency. The window size is generally chosen as 0.18 times the sampling frequency of 0.18 fs. For instance, with a sampling rate of 200 samples/s, the window width is set to 30 samples (150 MS).

The rising edge peak of the signal waveform obtained after moving window integration is marked as the main wave peak to be detected, and it is then adjusted using adaptive dual-threshold technology to determine the true main wave peak. If the peak value *DP* to be detected is greater than the threshold *T1*, it is the main wave peak value; otherwise, it is the noise peak value. The driver's heart rate signal extracted in the first 3 s is selected as the initial data, one-third of the maximum detected peak value is used as the initial signal threshold (*ST*), and half of the average value of all detected peak values is used as the initial noise threshold (*NT*). The adaptive dual threshold adjustment process is as follows:

If *DP* is the peak of the main wave:

If H is the main wave peak:

$$ST = \frac{1}{8}DP + \frac{7}{8}ST \tag{15}$$

If *DP* is the noise peak:

$$NT = \frac{1}{8}DP + \frac{7}{8}NT \tag{16}$$

Our dual thresholds, denoted as *T1* and *T2* for discrimination, vary with *ST* and *NT*. As *ST* and *NT* change, *T1* and *T2* dynamically adjust accordingly. This relationship can be expressed by the following formula:

$$T1 = NT + \frac{1}{4}(ST - NT) \tag{17}$$

$$T2 = \frac{1}{2}T1 \tag{18}$$

Considering the refractory period between two adjacent main waves and the physiological characteristics of the human heartbeat, we set the refractory period to 200 MS. During this period, redundant detection points are removed to prevent errors.

The average of the last eight DD intervals serves as the reference for the average interval. If the presently detected DD interval exceeds 1.66 times the average interval, indicating a potential detection miss, we initiate a backcheck using threshold *T2* and update the signal threshold as follows:

$$ST = \frac{1}{4}DP + \frac{3}{4}ST \tag{19}$$

### 2.2.3. Frequency Domain Analysis of Heart Rate Signals

Frequency domain analysis is used to depict the fundamental information regarding the changes in signal energy concerning frequency. The frequency domain component of the heart rate variability signal is intricately linked to the physiological state of the human body. Notably, high-frequency power mirrors the regulatory influence of the vagus nerve on the heart rate, while low-frequency power reflects the intricate interplay between sympathetic and parasympathetic nerves in the heart rate regulation process. The LF/HF ratio is a metric used to quantify the balance between sympathetic and parasympathetic tension. When the body is fatigued, sympathetic tension tends to dominate. Studies have indicated that the power spectral ratio of low-frequency power values (LF) and LF/HF to the heart rate variability signal significantly increases during fatigue, while the high-frequency power value diminishes. The LF/HF index serves as a crucial indicator of driver

sleepiness and fatigue status. Therefore, LF/HF is utilized as the frequency domain analysis index for the target signal. The specific algorithm flow is illustrated in Figure 5.

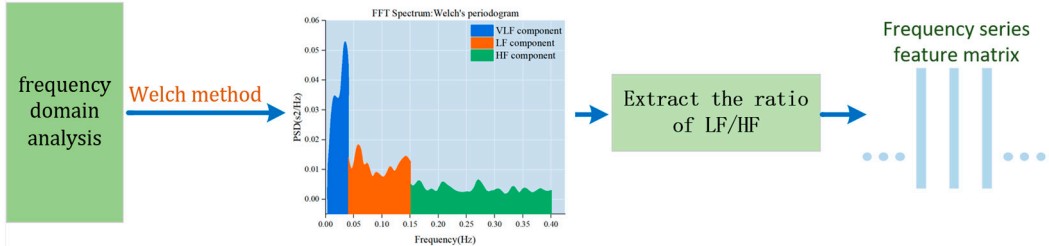

**Figure 5.** Frequency domain analysis of heart rate signals.

Given the target signal's inherent variability associated with a driver's heart rate and its substantial randomness, we use the Welch method to estimate the power spectrum of the target signal. The Welch method is an enhanced periodogram power spectral density estimation technique that is well-suited for rapid Fourier calculations. This method involves selecting window data, segmentally obtaining the power spectrum, and subsequently averaging it. The specific steps for our frequency domain analysis are outlined as follows.

First, we sample the obtained target signal to obtain the discretized signal $y(n)$ $(0 \leq n \leq N)$. The window size is positioned as $L$, and y(n) is divided into J segments when a quarter overlap is allowed, $J = (N - L/4)/(L/4)$. For the data in paragraph i:

$$y_i(m) = y[m + \frac{(i-1)L}{4}] \tag{20}$$

where $0 \leq m \leq L - 1, 1 \leq i \leq J$.

Consider the $i - th$ segment as an example to calculate the power spectrum of each segment of the data:

$$\hat{Y}_i(w) = \frac{1}{LU} |\sum_{m=0}^{L-1} y_i(m)D(m)e^{-jwm}|^2 \tag{21}$$

where, in this formula, $U = \frac{1}{L}\sum_{m=0}^{L-1} D^2(m)$ is the normalization factor, which ensures that the obtained spectrum is an asymptotically unbiased estimate, and $D(m)$ is the added window function. Next, we add the power spectra of all segments and take the average value to obtain the power spectrum y(n):

$$\widetilde{Y(w)} = \frac{1}{LUJ}\sum_{i=1}^{J} |\sum_{m=0}^{L-1} y_i(m)D(m)e^{-jwm}|^2 \tag{22}$$

The extracted time–frequency domain feature matrix is input into 1D CNN for processing. The 1D CNN method is effective in capturing the correlation between time–frequency domain features using convolution and pooling operations. HRV (heart rate variability) refers to the change in the heart rate over a period of time. The current HRV-based fatigue detection method typically obtains an ECG signal by attaching electrodes to the subject's skin and then converts the signal into HRV. However, obtaining HRV directly from the heart rate is not feasible. Since our goal is to learn how the heart rate signal changes over time, considering the heart rate of the sliding window over time can aid in achieving fatigue driving detection. Therefore, we designed a method to extract heart rate changes at adjacent moments during fatigue activities and established a 1D CNN-based model.

We utilize the 1D CNN for fatigue detection, as illustrated in Figure 6. The network comprises an input layer, three convolutional layers, and two fully connected layers. The input size is 1024 × 1. There are three convolutional layers with a filter length of 32, each utilizing the ReLU activation function. The convolution kernel sizes for layer 1, layer 2, and layer 3 are 16 × 1, 8 × 1, and 4 × 1, respectively. Following the convolutional layers, two fully connected layers (FCLs) with 256 and 128 neurons, respectively, are added for

classification. To prevent overfitting, a dropout layer is introduced after the fully connected layer. Finally, the SoftMax classifier calculates the probability of two fatigue states.

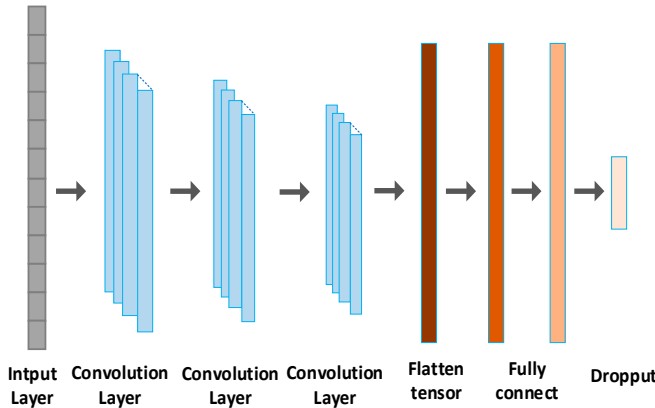

**Figure 6.** A 1D CNN based on fatigue detection.

### 2.2.4. Using 1D-MTCNN to Extract PERCOLS

Leveraging insights from [25], our approach acquires accurate key point coordinates with MTCNN. Subsequently, these key point coordinates are used to extract images of the eye and mouth regions. The extracted eye areas serve as input for the 1D CNN model to extract features. The detailed implementation process is depicted in Figure 7.

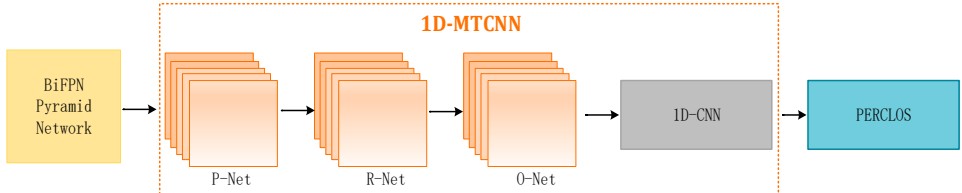

**Figure 7.** Using 1D-MTCNN to extract PERCOLS.

Local patterns and correlation information in sequence data can be captured using 1D CNN; therefore, it is suitable for processing video data from consecutive frames. The eye and mouth status of the region image is then classified. First, we connect the center points of the left and right eyes to obtain line *a* and define the angle between the connecting line *a* and the horizontal line as $\alpha$. The width and height of the eye area frame are defined as *w* and $h = w/2$, respectively. Then, we connect the left and right corners of the lips to obtain line *b* and draw a vertical line *b* from the key point of the nose to the connecting line *c*. The vertical distance is defined as *d*; then, the upper edge of the mouth area frame is $d/2$, and the lower edge is the vertical line *c*. At the extension line $5d/3$, after obtaining the eye and mouth area frames, we perform two classifications. With an interval of 60 s, there are a total of 1200 frames of images. Based on the PECLOS criterion and prior knowledge, we can state that:

$$P = \frac{\text{Eyes closed frames}}{\text{The total number of frames in the detection period}} \times 100\% \qquad (23)$$

$$L = \frac{\text{yawn frames}}{\text{The total number of frames in the detection period}} \times 100\% \qquad (24)$$

The PERCOLS algorithm has been proven to be able to accurately determine driver fatigue in real time. At the same time, based on prior knowledge, it can be determined whether the driver is in a fatigue state by detecting the number of times the subject yawns per minute. Therefore, the *p* value and the L value are selected as facial features.

Our proposed one-dimensional CNN architecture for fatigue detection is depicted in Figure 3. The architecture is composed of an input layer, three convolutional layers, and

two fully connected layers. The input size is specified as $600 \times 1$. The convolutional layers feature filter lengths of 24, using the ReLU activation function. The size of the convolution kernel is set at $10 \times 1$ for convolution layer 1, $5 \times 1$ for layer 2, and $3 \times 1$ for convolution layer 3. To finalize the network, two fully connected layers (FCLs) are added, containing 128 and 64 neurons, respectively, for classification. In order to mitigate overfitting, the SoftMax classifier is utilized to compute the probability of two fatigue states. The initialization of the network's values is accomplished by assigning random values. Given that PERCLOS and heart rate features are represented by one-dimensional signals, and considering the time and performance advantages of one-dimensional CNN in processing such signals, our choice of one-dimensional CNN for fatigue detection is well-founded.

### 2.3. Multi-modal Feature Fusion Fatigue Driving Identification Model

The LSTM network has shown unique advantages in the fields of text generation, machine translation, speech recognition, generated image description, and video tagging, demonstrating its powerful functions in processing and searching for spatio-temporal data. The schematic representation of the LSTM cell structure is shown in Figure 8.

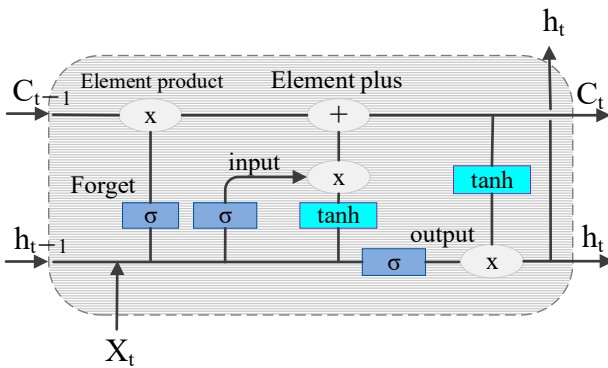

**Figure 8.** Schematic diagram of the LSTM cell structure.

The control gate mechanism of LSTM solves the problem of gradient disappearance when the RNN is processing a long sequence. The working principles of the three control gates are as follows:

The forget gate:

$$\Gamma_f = \sigma(W_f[h_{t-1}, x_t] + b_f) \tag{25}$$

the input gate:

$$\Gamma_u = \sigma(W_u[h_{t-1}, x_t] + b_u) \tag{26}$$

$$\widetilde{C}_t = \tanh(W_c[h_{t-1}, x_t] + b_c) \tag{27}$$

$$C_t = \Gamma u \times \widetilde{C}_t + \Gamma_f \times C_{t-1} \tag{28}$$

and the output gate:

$$\Gamma_o = \sigma(W_o[h_{t-1}, x_t] + b_o) \tag{29}$$

$$h_t = \Gamma_o \times C_t \tag{30}$$

where $x_t$ is the input at time $t$. $\Gamma_u, \Gamma_f, \Gamma_o$ are the input, forget, and output gates at time $t$, respectively. The output gate passes the activation function values $W_u, W_f, W_c, W_o$, and $b_u, b_f, b_o$ are the weights and deviations of the gates, respectively. $\widetilde{C}_t$ is the state of the memory element at time $t$, and $h_t$ is the final output.

A driver's driving state is a dynamic process, with the driver's physiological signal data changing over time, representing standard time series data. When utilizing LSTM to process the time–frequency domain time series feature matrix reflecting the driver's state, only past information is considered, and future information is disregarded [20], potentially impacting the accurate assessment of the current state. To address this limitation, Bi-LSTM emerges as a solution. Bi-LSTM comprises two LSTMs operating in opposite directions.

One LSTM processes information in a forward pass to retain past information, while the other LSTM processes information in a backward pass to incorporate future information. The outputs of these two LSTMs are then combined to derive the driver's status judgment using Softmax. The network structure of Bi-LSTM is illustrated in Figure 9. In driver fatigue detection, the time–frequency domain time series feature matrix reflecting the driver's state is fed into Bi-LSTM in real time. The forward and backward propagation layers in Bi-LSTM work together to precisely determine the driver's status by incorporating both past and future information.

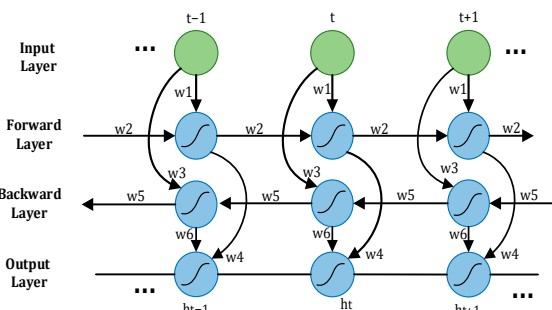

**Figure 9.** Bidirectional LSTM.

## 3. Experimental Results and Discussion

This section encompasses an introduction to the experimental environment, collected in-vehicle driving datasets, evaluation indicators, currently prevalent models, and a conclusive analysis of the experimental results. The experimental flow chart is depicted in Figure 10.

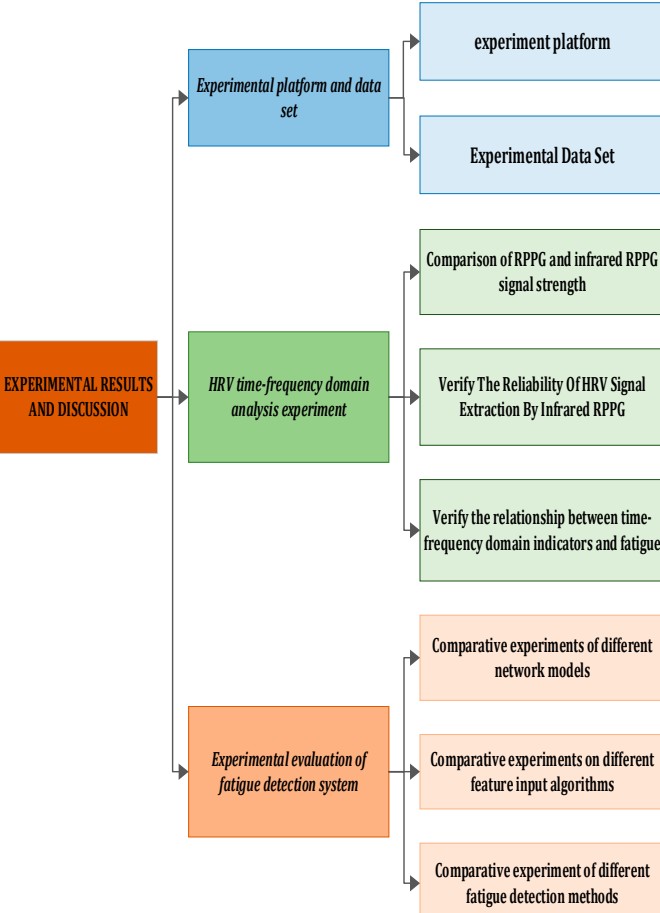

**Figure 10.** Experimental framework.

In this section, we elaborate on the details of the experiment. Figure 10 illustrates the entire experimental process, which is segmented into four parts. Initially, we introduce some fundamental experimental settings and datasets. Subsequently, utilizing the ECG signal as a reference, we measure the error induced by infrared rPPG, analyze heart rate variability, and calculate its correlation. Following that, the Bi-LSTM and LSTM networks are trained using time–frequency domain feature indicators, emphasizing the distinctive advantages of Bi-LSTM in detecting driver fatigue through comparison. Finally, we evaluate and analyze our model against existing physiological feature-based driver fatigue detection models.

### 3.1. Experiment Platform

#### 3.1.1. Experimental Equipment

The experimental equipment and environment include an infrared camera, Windows11 64-bit operating system, Intel i9 2.20 GHz processor, 16 GB memory, NVIDIA RTX 4060 (GPU), Python (3.7), and the Keras (Tensorflow2.1) framework.

In this research, the infrared camera serves as the pivotal hardware component for extracting the infrared rPPG signal. The original infrared time signal is derived from the video data collected with the infrared camera and utilized as the input for the rPPG signal. Consequently, the quality of the infrared camera profoundly influences the accuracy and reliability of heart rate signal extraction.

In selecting the appropriate infrared camera, several factors were taken into consideration, encompassing signal quality, performance, and cost. To guarantee the extraction of high-quality infrared rPPG signals, a cost-effective yet high-performance infrared camera with a superior signal-to-noise ratio was chosen, namely, the Oni S500 model. The Oni S500 infrared camera was selected for its advantageous cost-performance ratio and notable signal-to-noise ratio. This attribute proves pivotal in extracting delicate biological signals, as a high signal-to-noise ratio aids in diminishing interference and noise, ultimately enhancing the accuracy and stability of heart rate signal extraction.

The utilization of the Oni S500 infrared camera, known for its high performance and cost-effectiveness, allowed us to acquire high-quality infrared time signals for our research. This serves as a robust foundation for the subsequent processing and analysis of rPPG signals. Additionally, considering cost implications, the selection of the Oni S500 presents an economical hardware solution for our research, ensuring that this study can yield accurate and reliable experimental results.

#### 3.1.2. Experimental Dataset

In this study, due to the limited availability of public RGB and infrared multi-modal fatigue driving datasets, we opted to create our dataset, named the MDAD (Multi-Modal Driver Alertness Dataset). The dataset is illustrated in Figure 11. Our data collection used the Oni S500 binocular infrared camera, equipped with color RGB and infrared IR sensors. This binocular camera, capable of flexible installation on the rearview mirror or dashboard, simultaneously captures RGB and infrared images at a sampling rate of 30 Hz, with an image resolution of $640 \times 480$. Despite a minor offset in the positions of the RGB and IR cameras, it has been validated that this difference insignificantly impacts the effectiveness of data collection.

To ensure the diversity of the dataset, we collected real driving scenarios involving different types of vehicles (private cars, taxis, trucks, etc.) during both the daytime and nighttime. Throughout the data collection phase, a total of 52 participants (28 men and 24 women) with ages ranging from 22 to 53 years were involved. Each participant drove once on different road segments, engaging in typical driving activities. The entire driving session lasted approximately 2 h. Various real-world complexities, including changes in lighting conditions, driver head deflection, and partial occlusion, were intentionally introduced during the data collection to guarantee the diversity and randomness of the samples.

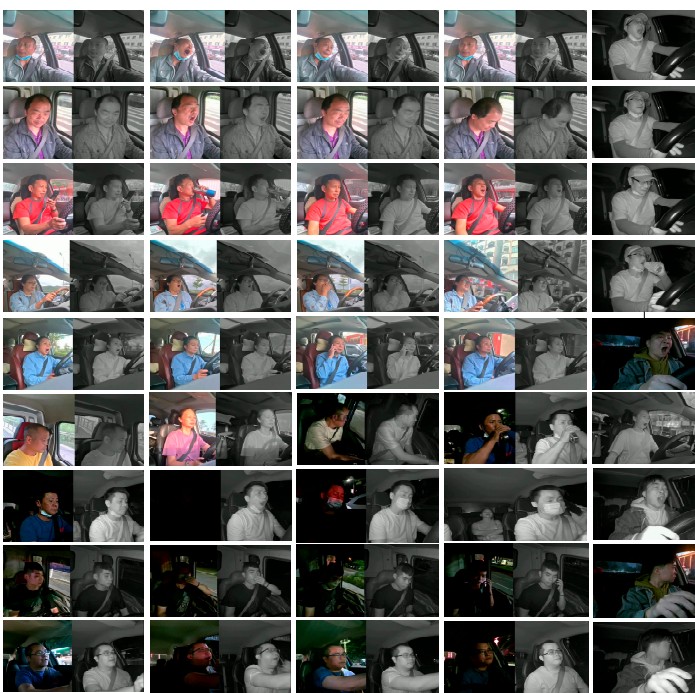

**Figure 11.** MDAD sample image.

Following the completion of data collection, we organized and segmented the videos into 60 s short video clips. These videos were then precisely labeled as either awake or fatigued based on the Karolinska Sleepiness Scale (KSS) in chronological order. The entire dataset comprises a total of 4000 1200 × 650 RGB-IR driving videos. Among these, 2530 are labeled as awake, while 1470 are labeled as fatigued.

The self-made MDAD serves as a crucial experimental foundation and resource for this research. Encompassing a diverse range of real driving scenarios, the dataset facilitates robust testing and evaluation of fatigue driving detection algorithms. This initiative contributes to enhancing the model's resilience and generalization. The utilization of this dataset is anticipated to drive further advancements in the field of fatigue driving detection.

*3.2. HRV Time–Frequency Domain Analysis Experiment*

In this section, we perform experiments to evaluate the accuracy and stability of our proposed HRV time–frequency domain analysis method for assessing driver fatigue. For the comparative analysis, we selected a widely used and advanced contact electrocardiogram (ECG) monitoring system. Fifteen participants, randomly chosen from a total of 52 subjects, were involved in the experiment. Subjects numbered 1–7 conducted the experiment at 12:00 noon, while subjects numbered 8–15 conducted it at 23:00 in the evening. This timing variation allowed us to accurately extract their heart rate (HR) and heart rate variability (HRV), including low frequency (LF) and high frequency (HF) data. Subjects were required to wear ECG monitors, ensuring correct placement for accurate signal extraction. The collected electrocardiogram and infrared video signals were then processed to extract heartbeat intervals for calculating heart rate variability.

3.2.1. Comparison of the RPPG and Infrared RPPG Signal Intensities

In the case of the long-distance extracted remote photoplethysmography (rPPG) signal, its signal strength is weakened due to the extended transmission distance, making it susceptible to environmental factors like lighting. The extraction of rPPG signals is crucial for our fatigue recognition task, particularly in relation to the extraction of heart rate variability (HRV) signals. If the obtained HRV signal is too weak or significantly affected by

noise interference, it can directly impact our subsequent data analysis, ultimately reducing the accuracy of driver fatigue assessment.

To mitigate this issue, we used both infrared rPPG and traditional rPPG methods for heart rate extraction. The experimental results, depicted in Figure 12, indicate that in comparison with the traditional rPPG signal, the heart rate signal obtained with infrared rPPG is less influenced by noise. Moreover, the heart rate signal acquired with infrared rPPG exhibits enhanced anti-interference capabilities. This implies that the rPPG signal, after minimizing environmental interference, is more robust, thus aiding in the extraction of accurate heart rate signals during subsequent data processing. Consequently, this contributes to an improved accuracy in assessing the driver's fatigue state.

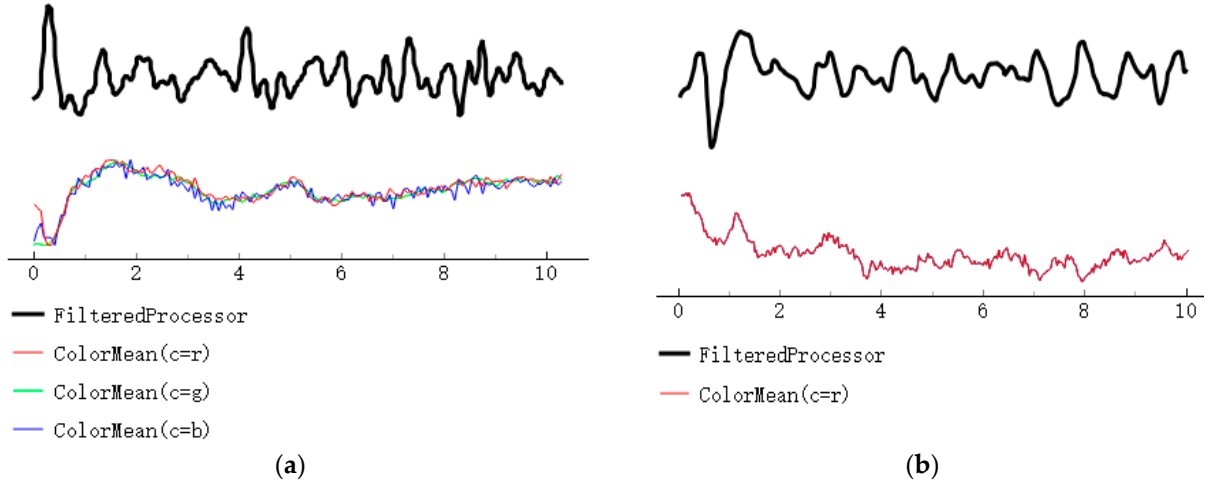

(**a**)                                                (**b**)

**Figure 12.** A comparison of the RPPG and infrared RPPG signal strengths. (**a**) Remote photoplethysmography; (**b**) Infrared remote photoplethysmography.

### 3.2.2. Verification of the Reliability of HRV Signal Extraction with Infrared RPPG

We assessed the reliability of HRV signal extraction using infrared rPPG by comparing the time–frequency domain index values of the ECG signal and the infrared rPPG signal, as illustrated in Figure 13. A comparison with the literature [16] reveals a high degree of overlap in the upper points of the polyline in Figure 13a,b for the SD value obtained from the infrared rPPG signal and the LF/HF ratio, which is consistent with the ECG signal. This suggests that the HRV signal can be effectively extracted from the infrared rPPG signal. Using electrocardiogram results as the standard, the accuracy of infrared rPPG, measured after 25 rounds of experiments, is approximately 95%.

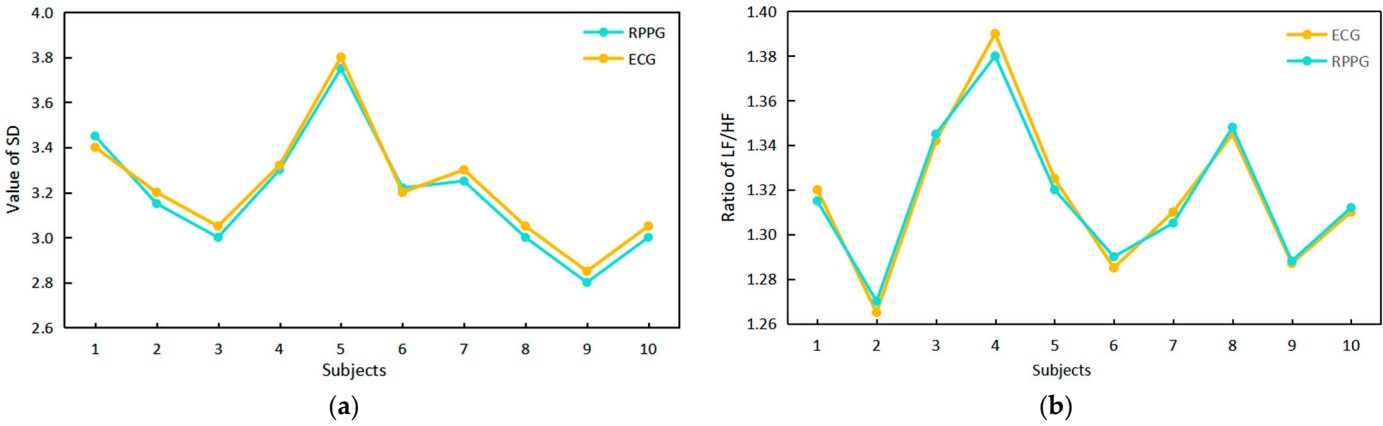

(**a**)                                                (**b**)

**Figure 13.** A comparison of the time–frequency domain index values of the ECG signals and infrared RPPG signals and a comparison of the RPPG and infrared RPPG signal intensities. (**a**) Time domain index comparison. (**b**) Frequency domain index comparison.

### 3.2.3. Verification of the Relationship between Time–Frequency Domain Indicators and Driver Fatigue

We conducted tests and verified, as indicated in the literature [14], that when transitioning from a normal state to a fatigue state, there are changes in time domain features such as SD and frequency domain features like the LF/HF ratio. This validates the feasibility of utilizing HRV time–frequency domain indices to detect driver fatigue.

The observations in Figure 14 indicate that when subjects transition from a normal state to a fatigue state, there are notable changes in both the SD value and the LF/HF value. With increasing fatigue, both SD and LF/HF values are higher than in the normal state, exhibiting a clear upward trend.

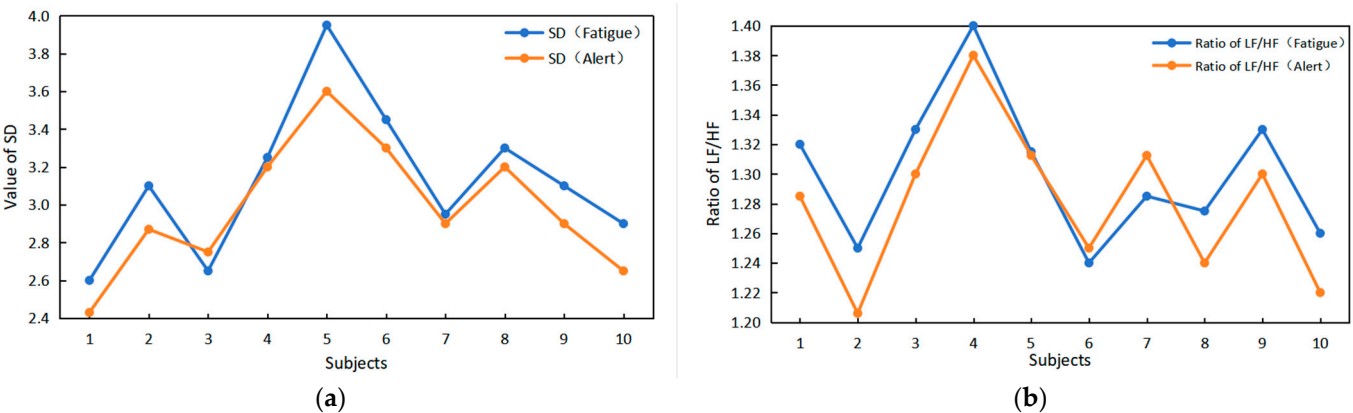

(**a**)          (**b**)

**Figure 14.** Verification of the relationship between time–frequency domain indicators and driver fatigue. (**a**) The value of SD changes between alarm and fatigue. (**b**) The LF/HF ratio changes between alarm and fatigue.

### 3.3. Experimental Evaluation of the Fatigue Driving Detection System

#### 3.3.1. Comparison of the Effects of BI-LSTM and LSTM Network Training

The experiment aimed to assess the training effect of the Bi-LSTM network. Thirty sets of 30 s facial videos were utilized to train both the Bi-LSTM and LSTM network models. A comparison of the loss function and accuracy between the two networks is presented in Figure 10. The results reveal that in most cases, Bi-LSTM outperforms LSTM, demonstrating superiority in fatigue detection. This suggests that integrating both past and future physiological signals of the driver enhances the judgment process, leading to improved results. As shown in Figure 15, after the 15th training cycle, the loss function approaches 0, and around the 20th cycle, the accuracy of Bi-LSTM approaches 98%.

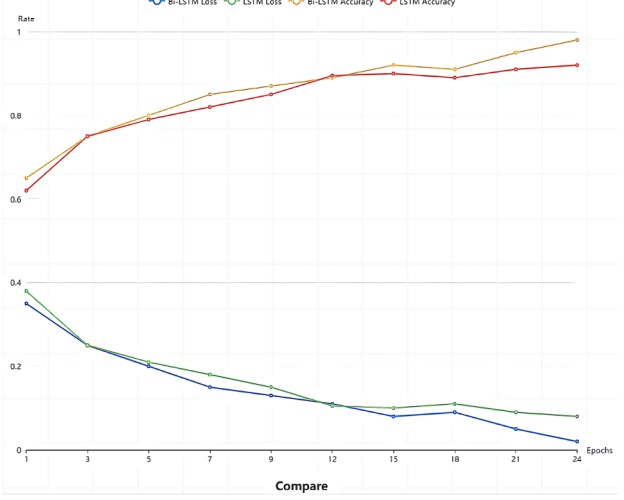

**Figure 15.** Comparison of the LSTM and Bi-LSTM models.

### 3.3.2. Model Evaluation

The sensitivity (*Se*), true positive rate (+*P*), and accuracy rate (*Acc*) were used to evaluate the overall performance of the proposed algorithm. The calculation formulas are as follows:

$$Se = \frac{TP}{TP + FN} \tag{31}$$

$$+P = \frac{TP}{TP + FP} \tag{32}$$

$$Acc = \frac{TP}{TP + FP + FN} \tag{33}$$

where true positive (*TP*) represents the number of correctly detected fatigue states or non-fatigue states, false positive (*FP*) represents the number of non-fatigue states judged as fatigue states, and false negative (*FN*) indicates the number of fatigue states judged as non-fatigue.

The average absolute error is the average of the sum of the absolute differences between the *DD* interval values of all test samples in the test sample and the *RR* interval control value, which can be expressed as:

$$\overline{Error} = \frac{1}{N} \sum_{i=1}^{N} |DD_i - RR_i| \tag{34}$$

where *N* represents the total number of test samples.

### 3.3.3. Comparative Experiments on Different Modal Feature Input Algorithms

By comparing the literature [34], this study aims to more effectively evaluate the classification effect after multi-modal data fusion and the single-modal classification effect. To ensure consistency in the experimental environment and subjects, different modal features were input into various algorithms for comparison. The feature categories were divided into four groups: physiological features, visual features, mixed visual features, and mixed features. These four groups of feature types were categorized into a fatigue state and an awake state based on given labels. They were then input into traditional SVM, DBN, random forest, and CNN networks simultaneously to compare the obtained classification accuracy with the method proposed in this paper. The experimental results are shown in Table 1.

**Table 1.** Comparison of the accuracy rates of different algorithms input with different feature types.

| Category | Feature | Index | SVM | Random Forest | DBN | CNN | Ours |
|---|---|---|---|---|---|---|---|
| Physiological characteristics | ECG<br>rPPG | HR, HRV<br>HRV | 0.792<br>0.778 | 0.827<br>0.817 | 0.853<br>0.827 | 0.874<br>0.864 | **0.894**<br>**0.895** |
| Visual characteristics | Eyes<br>Mouth | PERCLOS<br>Yawn frequency | 0.785<br>0.757 | 0.822<br>0.783 | 0.849<br>0.806 | 0.868<br>0.855 | **0.877**<br>**0.895** |
| Mixed visual features | Eyes + Mouth | PERCLOS, yawn frequency | 0.814 | 0.836 | 0.869 | 0.896 | **0.948** |
| Mixing multi-class features | Eyes + Mouth + rPPG | PERCLOS, yawn frequency, HRV | 0.774 | 0.842 | 0.871 | 0.949 | **0.972** |

Note: Bold is the best result.

According to Table 1, the proposed method in this paper achieves the highest classification accuracy across all feature types. Concerning physiological characteristics, our method demonstrates relatively high accuracy rates when utilizing both ECG and rPPG, achieving 0.894 and 0.895, respectively. Regarding visual features, the accuracy of our method using eye and mouth features is 0.877 and 0.895, respectively, also surpassing the traditional classification algorithms. Notably, the combination of visual features (Eyes + Mouth) performs

exceptionally well in our method, achieving an accuracy of 0.948, significantly outperforming other algorithms. Additionally, we observe that compared with traditional algorithms (SVM and random forest), the deep learning models (DBN and CNN) exhibit superior performance, affirming the advantages of deep learning in processing multi-modal data. In the case of mixed multi-class features (Eyes + Mouth + rPPG), our method excels with an accuracy of 0.972, surpassing the other traditional algorithms.

In summary, based on the data results from the comparative experiments, we can confidently conclude that the multi-modal fusion fatigue detection method proposed in this article demonstrates high accuracy across various feature types. The optimal performance is achieved when combining multiple types of features, establishing the effectiveness and promising application prospects of this method in fatigue detection.

### 3.3.4. Comparative Experiments on Different Fatigue Driving Detection Methods

In order to enhance the generalization and reliability of the experimental results, we conducted a comparison between the algorithm proposed in this paper and the existing mainstream fatigue driving detection algorithms. This comparison includes the multi-class support vector machine (MCSVM) presented in [20] and the multi-granularity deep convolution model (RF-DCM) introduced in [27]. The dataset utilized for experimentation is our self-compiled MDAD, comprising continuous facial videos of drivers navigating diverse and challenging driving scenarios. The accuracy of each algorithm in fatigue detection was evaluated, and the results are presented in Table 2 below.

**Table 2.** Comparison of the accuracy of fatigue driving detection methods in complex environments.

| Method | Acc (%) |
| --- | --- |
| MCSVM [36] | 87.3% |
| RF-DCM [37] | 94.6% |
| Drowsiness detection system [28] | 85% |
| Fatigue-Radio [33] | 81.4% |
| FFD system [31] | 97.8% |
| MTCNN + InceptionV3 [10] | 91.1% |
| HDDD [38] | 88.9% |
| CNN + DF-LSTM [39] | 88.9% |
| Fusion system [32] | 97.3% |
| Ours | **98.2%** |

Note: Bold is the best result.

## 4. Conclusions

To address the challenges associated with the low accuracy and poor robustness of traditional fatigue detection methods, particularly under varying lighting conditions, we propose a non-invasive approach for multi-modal fusion fatigue detection that integrates heart rate features with eye and face features. Leveraging an infrared camera in conjunction with rPPG and MTCNN combined with the BiFPN pyramid network, we extract a driver's physiological characteristics and facial features, mitigating errors introduced by day-to-night lighting changes and enhancing the stability and reliability of fatigue detection. Furthermore, we introduce a feature extraction enhancement module based on an improved Pan–Tompkins algorithm and 1D-MTCNN to more accurately extract heart rate signals and eyelid features. These enhancement modules contribute to elevating the quality and precision of the data, forming a solid foundation for subsequent fatigue detection models. Two independent fatigue detection models are established using a one-dimensional convolutional neural network (1D CNN), where one model is based on the PERCLOS value, and the other is based on the heart rate signal. After training these models, accurate detection and classification of different fatigue characteristics are achieved. To enhance robustness, real-time performance, and accuracy, the BiLSTM network is used to input the data results from the two trained models, allowing it to learn the temporal relationships between the data. This dynamic modeling approach effectively processes

forecast time series input data, improves the model's applicability in real-world scenarios, and realizes multi-modal fatigue detection. Extensive experiments and comparisons of the self-compiled MDAD demonstrate that the proposed model exhibits excellent fatigue detection performance, boasting high accuracy and robustness across different scenarios. The multi-modal fusion fatigue detection method presented in this paper provides an effective solution for achieving accurate and reliable fatigue driving detection.

**Author Contributions:** Conceptualization, L.K. and K.X.; methodology, L.K.; software, K.N.; validation, K.N.; formal analysis, L.K.; investigation, J.H.; resources.; data curation, K.X.; writing—original draft preparation, L.K.; writing—review and editing, K.N.; visualization, J.H.; supervision, K.X.; project administration, W.Z.; funding acquisition, K.X. All authors have read and agreed to the published version of the manuscript.

**Funding:** This work was supported by the National Natural Science Foundation of China (No. 62373372 and 62272485), under whose cooperation, this work was successfully established and conducted. This work was mainly supported by a project sponsored by the Undergraduate Training Programs for Innovation and Entrepreneurship of Yangtze University under Grant Yz2022062. This project enabled us to purchase and build a non-invasive fatigue detection system based on multi-modal fusion and conduct the experiments in this paper to complete the identification of fatigue states under different lighting conditions.

**Institutional Review Board Statement:** Not applicable. The digital products mentioned in this article are for illustrative and explanatory purposes only and do not imply any advertising or marketing intent.

**Informed Consent Statement:** Informed consent was obtained from all subjects involved in this study.

**Data Availability Statement:** The original contributions presented in the study are included in the article, further inquiries can be directed to the corresponding author.

**Acknowledgments:** We express our sincere gratitude to all the drivers who willingly participated in the dataset collection. Their support and cooperation have been invaluable to the success of this research.

**Conflicts of Interest:** The authors declare no conflicts of interest.

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
