# Peer review of "Remote Photoplethysmography and Motion Tracking Convolutional Neural Network with Bidirectional Long Short-Term Memory: Non-Invasive Fatigue Detection Method Based on Multi-Modal Fusion"

_sensors, doi:10.3390/s24020455_

Round 1
Reviewer 1 Report
Comments and Suggestions for Authors
In this article, a fatigue detection method is proposed from the perspective of multi-modal fusion. The content of this article is rich. However, the quality and impact of the work are limited. The paper could be strengthened in the following aspects:
1. The article uses the Karolinska Sleepiness Scale (KSS) for fatigue assessment. However, the scale is highly subjective and may lead to inaccurate labeling of fatigue status. Please clarify how this potential inaccuracy is mitigated.
2. The article employs neural networks and machine learning models, yet the method for partitioning the training and prediction sets is not clearly described. It is also unclear whether the training and test data are derived from the same subjects. Please clarify these aspects to ensure the robustness of the analysis.
3. The entire article focuses on the extraction of human physiological signals, such as heart rate and features of HRV, utilizing the rPPG method. However, in subsection 3.2, it is mentioned that the ECG signals of the subjects are acquired using contact ECG equipment for physiological feature extraction. This raises a contradiction, as if the ECG signals are obtained via contact ECG equipment, what is the relevance of the proposed rPPG method for physiological signal extraction? Please clarify this point to avoid any confusion.
4. In this paper, 52 experimental subjects were recruited, yet in subsection 3.2.2, when assessing the reliability of IR rPPG heart rate variability signal extraction, only 10 groups of subjects were utilized for verification. This relatively small sample size may limit the generalizability of the findings. Please consider increasing the sample size for more robust results.
5. In subsection 3.1.2, it is mentioned that the acquired video is segmented into short 60-second videos. However, in subsection 3.3.1, a 30-second facial video is utilized to train the Bi-LSTM and LSTM models. This raises a question regarding the applicability of the obtained optimal model to the shorter 60-second videos. Please clarify this aspect to ensure consistency.
Author Response
Please see the attached file:Response to Reviewers 1.docx.

Reviewer 2 Report
Comments and Suggestions for Authors
The paper titled "RPPMT-CNN-BiLSTM: Non-invasive Fatigue Detection Method Based on Multi-modal Fusion" proposes a novel approach to detecting driver fatigue using non-invasive multi-modal fusion techniques. Here are some technical comments and suggestions for potential improvements:
The paper introduces a unique combination of technologies for fatigue detection, including the RPPMT-CNN feature extraction enhancement network and BiLSTM for dynamic modeling. This multi-modal approach, integrating both physiological and facial features, represents a significant advancement in non-invasive fatigue detection technologies​​.
The utilization of non-contact multi-modal feature extraction methods, including the use of the Haar-Cascade face detector and SLIC algorithm, is commendable. This approach ensures accurate identification of the driver's physiological signals and facial features, minimizing interference from motion artifacts and lighting changes​​. However, the paper could benefit from further exploration of the impact of extreme environmental conditions, such as very low light or high dynamic range scenarios, on the efficacy of these feature extraction methods.
The method for processing physiological signals, particularly the use of remote Photoplethysmography (rPPG), is innovative. This non-invasive technique enhances user comfort and practicality for real-time monitoring. The paper could be strengthened by providing more details on the reliability of rPPG signals in varying conditions and its comparison with traditional PPG methods​​.
The use of the enhanced Pan-Tompkins algorithm for primary wave detection and the implementation of singular spectrum analysis (SSA) for signal separation are notable advancements. These techniques aid in accurately identifying heart rate signals and reducing noise interference. Further empirical validation of these enhancements, particularly in comparison with existing methods, would substantiate the claimed improvements​​.
Author Response
Please see the attached file:Response to Reviewers 2.docx.

Reviewer 3 Report
Comments and Suggestions for Authors
The paper introduces a non-invasive fatigue detection method, RPPMT-CNN-BiLSTM, which incorporates multi-modal fusion. The combination of heart rate signals and PERCLOS values, along with the use of 1D neural networks and BiLSTM, represents a comprehensive and innovative approach to fatigue detection. The inclusion of a feature extraction enhancement module based on the improved Pan-Tompkins algorithm and 1D-MTCNN is a notable strength. This enhancement addresses challenges related to the accuracy of heart rate signal extraction and eyelid features, improving the reliability of the proposed method. The integration of BiLSTM for dynamic modeling adds a temporal aspect to the fatigue detection, allowing for a more nuanced understanding of the driver's state over time. This contributes to the robustness of the proposed method. The achieved accuracy of 98.2% on the self-made MDAD dataset is impressive and indicates the effectiveness of the proposed method. High accuracy is crucial for real-world applications, particularly in the context of smart transportation.
Here are some suggestions for improvement:
While the paper provides an overview of the proposed method, certain sections could benefit from more detailed explanations. Consider providing step-by-step details of the algorithm, especially in the feature extraction enhancement module and the dynamic modeling with BiLSTM. Besides, the manuscript will benefit from adding more relevant work on the application of hybrid CNN and BiLSTM, such as: doi.org/10.1007/s40436-023-00464-y, and doi.org/10.3390/electronics12010232.
To demonstrate the significance of the proposed method, consider including a comparative analysis with existing fatigue detection methods. Highlight how the RPPMT-CNN-BiLSTM approach outperforms or complements other methods in the literature. Discuss any limitations of the proposed method. This could include scenarios where the method might not perform as well or challenges associated with real-world implementation.
The paper presents a robust and innovative approach to non-invasive fatigue detection through multi-modal fusion. With more detailed explanations, a comparative analysis with existing methods, a discussion of limitations, and considerations for ethical implications, the paper could further solidify its contribution to the field of smart transportation and driver monitoring.
Comments on the Quality of English LanguageThe quality of English in the manuscript is generally good, with clear and coherent language.
Author Response
Please see the attached file:Response to Reviewers 3.docx.
